# High-frequency vs. low-frequency MIDI-assisted group music therapy in psychiatric inpatients: A randomized controlled trial

Manuel Esteban-Cárdenas[1], Ana Gómez-Puentes[2,3], Carlos Torres-Delgado[3], Adrián Hidalgo-Valbuena[4], Eugenio Ferro [1,3]*

1 Graduate School of Psychiatry, Universidad El Bosque, Bogotá, Colombia, 2 Department of Epidemiology, Universidad El Bosque, Bogotá, Colombia, 3 Department of Research, Instituto Colombiano del Sistema Nervioso - Clínica Montserrat, Bogotá, Colombia, 4 Occupational & Music Therapy Unit, Instituto Colombiano del Sistema Nervioso - Clínica Montserrat, Bogotá, Colombia

* eferro@unbosque.edu.co

## Abstract

### Background

Music therapy has been increasingly incorporated into psychiatric inpatient care as an adjunctive intervention; however, evidence regarding its comparative effectiveness, particularly with respect to intervention intensity during brief psychiatric hospitalizations, remains limited. Most randomized trials have evaluated interventions delivered over several weeks, leaving uncertainty about the potential differential effects of higher versus lower intervention intensity within short inpatient stays.

### Methods

This randomized controlled trial with a parallel-group design was conducted in an adult psychiatric inpatient unit in Bogotá, Colombia. Participants were randomly assigned (1:1) to either a high-frequency music therapy group, which received five sessions delivered on consecutive days within a single inpatient week, or a low-frequency group, which received one session during the same period. Allocation concealment was ensured through computer-generated randomization performed by an independent researcher not involved in participant recruitment or outcome assessment.

Primary outcomes were symptoms of depression, anxiety, and stress, assessed using the Depression Anxiety Stress Scale–21 (DASS-21) at baseline and at the end of the intervention week. Secondary outcomes included emotional responses to music therapy, assessed using selected items from the Questionnaire of the Impact of Music Therapy Sessions in Adults (CISMA instrument), and global life satisfaction, measured with the Satisfaction With Life Scale (SWLS). Between-group changes over time were analyzed using repeated-measures mixed-effects linear models.

**Data availability statement:** All data files are available from the OSF Storage database (Ferro, E. 2024, October 15). High-Frequency vs Low-Frequency Music Therapy in Psychiatric Inpatients: A Randomized Controlled Trial. https://doi.org/10.17605/OSF.IO/384FB.

**Funding:** The author(s) received no specific funding for this work.

**Competing interests:** The authors have declared that no competing interests exist.

## Results

A total of 91 participants were randomized, of whom 74 (37 per group) completed post-intervention assessments and were included in the per-protocol analysis. Between-group analyses revealed a statistically significant group × time interaction for the DASS-21 stress subscale only. Participants in the high-frequency music therapy group showed a greater reduction in stress scores compared with the low-frequency group, with a statistical between-group difference (6.49 ± 5.55 vs. 7.03 ± 5.81; p = 0.023).

No statistically significant between-group differences were observed for the DASS-21 anxiety subscale (p = 0.339) or the depression subscale (p = 0.270). Global life satisfaction, as measured by the SWLS, did not differ significantly between groups (p > 0.05). For secondary outcomes assessed using selected CISMA items related to emotional change, between-group differences were small and not statistically significant across all items. Overall, except for stress symptoms, outcomes did not differ significantly between the high-frequency and low-frequency intervention groups.

## Conclusions

In this randomized controlled trial, a higher frequency of music therapy sessions was associated with a greater reduction in stress symptoms compared with a lower-frequency intervention during a brief psychiatric hospitalization. However, no between-group differences were observed for anxiety, depression, life satisfaction, or emotional response measures. Overall, these findings indicate limited evidence of differential effectiveness by intervention intensity and should be interpreted cautiously in light of the study's methodological limitations.

## Trial registration

ISRCTN registry: ISRCTN87861817 (https://www.isrctn.com/ISRCTN87861817).

## Introduction

The relationship between music and health has been widely studied [1]. It has evolved as a complementary intervention for the treatment of mental illnesses and disorders. The American Music Therapy Association (AMTA) defines music therapy as the clinical and evidence-based use of musical interventions to achieve specific therapeutic goals in the treatment of various physical and psychological disorders [2]. They are based on the premise that music influences emotional, cognitive, and physical aspects of human well-being, stimulating areas of the brain linked to emotions, memory, and behavior [2].

Music therapy, in addition to demonstrating benefits for people's general well-being, is used as a complementary therapeutic intervention in the treatment of various physical and mental health conditions [3,4]. Its versatility allows it to be applied in different settings, populations, and pathologies [5–7].

It has been demonstrated that music therapy produces modulation of neurochemical systems, promoting relaxation and well-being [6]. In addition, it promotes neuroplasticity and the release of neurotransmitters such as dopamine and serotonin, which are related to pleasure, motivation, and well-being [8]. Both passive and active participation in its creation have shown positive effects on cognitive and psychosocial functioning [5] as well as reducing pain and anxiety [9].

Previous studies support music therapy as an effective non-pharmacological intervention for emotional regulation, significantly reducing symptoms of anxiety and depression in different populations, including patients with chronic diseases such as those on hemodialysis [10], people with dementia [11], and psychiatric patients, highlighting its impact on improving mood and self-expression [12]. It has also been reported that the combination with Progressive Muscle Relaxation (PMR) is effective in reducing anxiety, depression, and stress in patients with breast cancer and gynecological cancer during chemotherapy, also improving the life satisfaction of these patients [13].

In acute psychiatric settings, music therapy has shown particular promise in enhancing emotional expression, social interaction, and patient engagement. A systematic review by Carr summarizing 98 studies found that music therapy in psychiatric inpatient units tends to emphasize active, non-verbal expression through improvisation and structured musical activities, often complemented by verbal reflection [14]. These interventions have been associated with reductions in anxiety, depression, stress and improvements in patient communication and emotional regulation, despite variability in methodologies and limited sample sizes [14].

Contemporary evidence supports the use of music therapy as an effective intervention in acute psychiatric settings, even within contexts of brief hospitalization [15]. Clinical trials and feasibility studies have shown that music therapy delivered during acute episodes is associated with improvements in psychiatric symptoms, emotional regulation, motivation, and treatment adherence, as well as increased therapeutic engagement in inpatient units. Systematic reviews and meta-analyses have consistently confirmed a dose–response relationship, whereby greater therapeutic exposure is associated with larger clinical effects [16]. Nevertheless, recent evidence suggests that short-duration interventions may elicit rapid and clinically meaningful responses, particularly in the reduction of depressive symptoms and in strengthening early treatment engagement [17]. These findings are especially relevant for acute psychiatric care, where limited lengths of stay necessitate structured, intensive interventions capable of producing early therapeutic benefits.

However, although music therapy has demonstrated its effectiveness, important questions remain regarding the optimal frequency and duration of the intervention, particularly in the context of acute psychiatric hospitalization. Most studies in the literature are based on longer-term interventions with weekly sessions and follow-up periods that are not easily applicable in inpatient settings, where hospital stays have become increasingly short, typically lasting 10–12 days [18]. This makes it challenging to implement and evaluate interventions that require sustained engagement over time. Consequently, it remains unclear what level of intensity or frequency is most appropriate and effective in these acute care contexts. Our study seeks to explore this question by assessing the feasibility and potential benefits of a short-term, intensive music therapy approach.

## Materials and methods

### Study design

This study is a randomized controlled clinical trial with a parallel group design (the Clinical Study Registry is available at the ISRCTN registry under registration number: ISRCTN87861817). Participants were randomly assigned to either the high-frequency music therapy intervention group or the low-frequency control group. The high-frequency intervention group received five sessions of music therapy in one week of inpatient treatment at a rate of one session per day for five consecutive days. The low-frequency control group received one music therapy session in one week of inpatient treatment.

The primary outcome was symptoms of depression, anxiety, and stress during inpatient treatment with the Depression/Anxiety/Stress Scale of 21 items (DASS-21). Secondary outcomes were the impact of music therapy sessions on adult patients measured with the CISMA questionnaire (CISMA by Spanish acronyms) and global life satisfaction.

## Participants

Participants were receiving inpatient treatment at a tertiary referral medical center for psychiatry in Bogotá, Colombia (ICSN – Clínica Montserrat – Hospital Universitario) during their participation in the study.

Inclusion criteria were: 1) individuals over 18 years of age; 2) more than 48 hours of psychiatric hospitalization at the time of recruitment; 3) pharmacological treatment with medication adjustments in the last two weeks. Exclusion criteria were: 1) previous participation in music therapy programs; 2) main diagnosis of hospitalization, abstinence syndrome, or substance dependence; and 3) more than six days of hospitalization at the time of selection.

Patients were recruited consecutively between August 1 and September 2, 2024, upon admission to the psychiatric inpatient unit, until the target sample size was reached. All interventions, including follow-up assessments, were completed by September 9, 2024. Participant selection was conducted after 3–5 days of hospitalization, once clinical stability was determined by the treating physician to ensure the patient could complete the music therapy intervention. A total of 106 patients were assessed for eligibility during this period. Fifteen declined to participate, resulting in 91 patients who provided written informed consent and were enrolled in the study.

Participants were randomly assigned to one of two groups: the high-frequency intervention group (5 sessions/week) or the low-frequency control group (1 session/week). Of the 91 enrolled participants, 2 in the intervention group and 1 in the control group were lost before completing the first CISMA and SWLS assessments, leaving 45 and 46 participants, respectively. During the second follow-up, 6 additional participants from the high-frequency group and 8 from the low-frequency group were lost, resulting in 37 participants in each group completing the second CISMA and SWLS assessments as well as the post-intervention DASS-21 (Fig 1). All participants remained in their originally assigned groups throughout the study, with no crossover or reassignment, thereby preserving the integrity of the randomized design.

Cohorts of recruitment were formed approximately every 4 days based on admission date, resulting in eight sequential recruitment periods during the study enrollment phase (1 August to 2 September). Participants of the high-frequency group (n = 37) who did not complete all five intervention sessions were excluded from the analysis. All participants in the control group (n = 37) received one music therapy session and were allowed to attend additional sessions thereafter; however, no further outcome measures were collected from them for analysis.

Assessments were conducted at multiple time points. The DASS-21 and the Satisfaction With Life Scale (SWLS) were administered twice: at baseline (before the first music therapy session) and after the fifth day of participation. The CISMA scale was applied in a pre-post format during the first music therapy session for all participants, and additionally during the fifth and final session for those in the high-frequency group. Patients who were discharged before completing post-intervention assessments were excluded from the final analysis.

## Sample size calculation

The sample size calculation was performed using RStudio (version 2024.09.0 + 375), employing a two-sample t-test model with a one-tailed hypothesis, assuming a significance level of α = 0.05, a statistical power of 80% (1 − β = 0.80), and a 1:1 allocation ratio between groups. Based on the findings of Zhang et al. [11], we expected a directional effect, namely that the high-frequency music therapy group would achieve a greater reduction in psychological distress than the low-frequency group. Under this assumption of a medium effect size (Cohen's d = 0.6), approximately 36 participants per group would be required. Considering a potential attrition rate of 20%, we planned to enroll a total of 74 participants (37 per group) to ensure adequate statistical power.

We acknowledge that the a priori sample size calculation was based on a simple two-sample t test, whereas the final analysis employed a repeated-measures mixed-effects model. Power estimation for linear mixed models is considerably more complex and requires specifying variance components and within-subject correlations that were not available at the protocol development stage. Consequently, a simplified two-sample t-test approximation was used for the initial

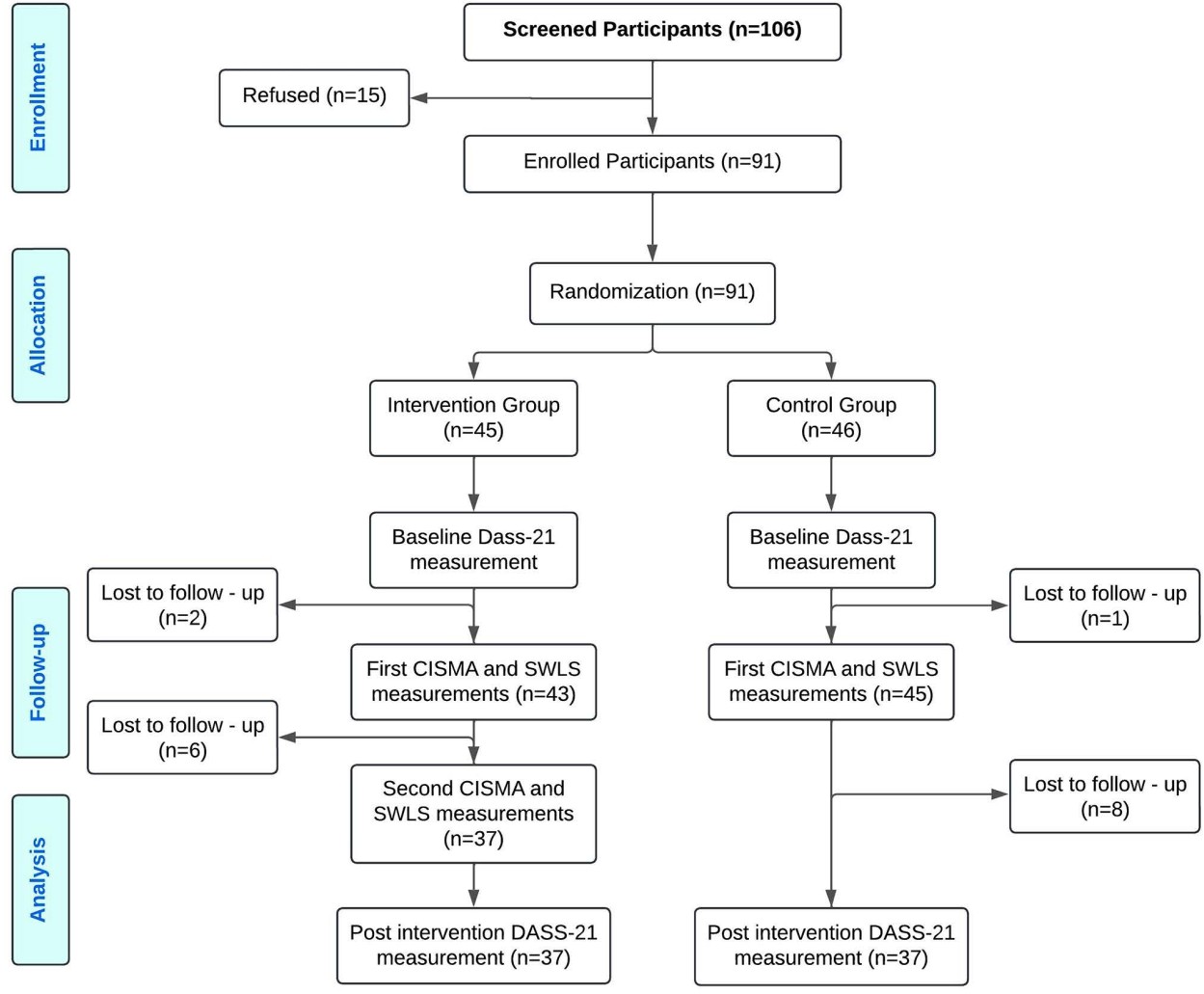

**Fig 1. Flow chart of study participation.** The number of patients recruited and how many completed each study phase is shown. Source: Own elaboration.

estimation. Therefore, the calculated sample size should be interpreted as an approximation, and the study may have been underpowered to detect group-by-time interaction effects.

Sampling was non-probabilistic and based on convenience. The intervention was designed to be implemented over a short period, given the brief duration of patients' hospital stays.

### Randomization and allocation concealment

Participants were then randomly assigned in a 1:1 ratio to either the high-frequency music therapy intervention group or the low-frequency control group.

The random allocation sequence was generated using a computer-based random number generator to ensure an equal probability of group assignment. The allocation sequence was concealed from the study coordinator responsible for participant recruitment until the moment of assignment. The study employed simple randomization, ensuring that each

participant had an equal probability of being assigned to either group. No restrictions, such as blocking or stratification, were applied during the randomization process.

## Blinding

Due to the nature of the music therapy intervention, blinding of participants and care providers was not feasible. Participants were aware of the frequency of music therapy sessions assigned, and the music therapist delivering the intervention was not blinded to group allocation. However, participant recruitment and enrollment were conducted by a study coordinator who was blinded to the random allocation sequence. The randomization list was generated independently and was not accessible to the recruiter at the time of enrollment. Outcomes were assessed using self-reported questionnaires, and no formal blinding of outcome assessment was implemented. These methodological aspects were considered in the interpretation of the results.

## Intervention

This intervention was developed to promote emotional regulation, sustained attention, creativity, and group cohesion among psychiatric inpatients. It was grounded in principles of active music therapy and incorporated progressive muscle relaxation and multisensory stimulation to enhance therapeutic engagement. The goal was to foster new sensory and creative experiences that could reduce catastrophic thinking and enhance the sense of awe and presence.

Sessions were conducted in a controlled indoor therapeutic space at a psychiatric ward and followed a structured four-phase format: warm-up, group synchronization, main activity, and closure. During the warm-up, a standardized welcome song promoted group cohesion and emotional safety. The second phase, group synchronization, incorporated progressive muscle relaxation, guiding participants through breathing and muscle tension release to prepare them for the creative activities. The third phase, main activity, introduced the Playtron MIDI controller, which allowed participants to interact musically with conductive objects such as fruits, water, and plants, each assigned a unique sound via a mobile application. This playful and novel engagement enhanced sustained attention, group role-play, creativity, and sensory exploration. Traditional instruments, including a portable piano and guitar, were also used to support the musical process. Finally, the closing phase featured a farewell song and an optional ritual where participants ate the fruit used during the session, reinforcing a sense of ownership, emotional integration and group cohesion.

The intervention was led by a professional music therapist with a master's degree, supported by a physician trained in psychological first aid and a psychiatry resident. This multidisciplinary team supervised the sessions to ensure both clinical safety and adherence to the protocol. Although no formal fidelity assessment tool was used, adherence was informally monitored through session observations and team supervision, maintaining consistency across all groups.

The intervention was delivered through group-based, in-person sessions held in a controlled therapeutic environment. Each session lasted 30 minutes and accommodated 8–16 participants per group for the creation of musical compositions in ensembles through the use of conductive objects (like fruits, water, and plants), aiming to engage them creatively and sensorially. Traditional musical instruments such as piano and guitar were used alongside a Playtron MIDI controller, which was connected to conductive objects. Each object was assigned to a unique sound via a mobile application, encouraging collaborative music-making and enhancing both individual creativity and group expression (Fig 2).

The music therapy intervention was designed and delivered as a complex, multicomponent intervention. To ensure transparency and reproducibility, the intervention is reported in accordance with the Template for Intervention Description and Replication (TIDieR) checklist. A detailed description of the intervention components, materials, procedures, provider qualifications, mode of delivery, session frequency and duration, tailoring, and fidelity considerations is provided in **Supplementary Appendix [**S1 Appendix**]**. The description in the present section summarizes the core elements of the intervention, while the full TIDieR-based reporting is available in the supplementary material.

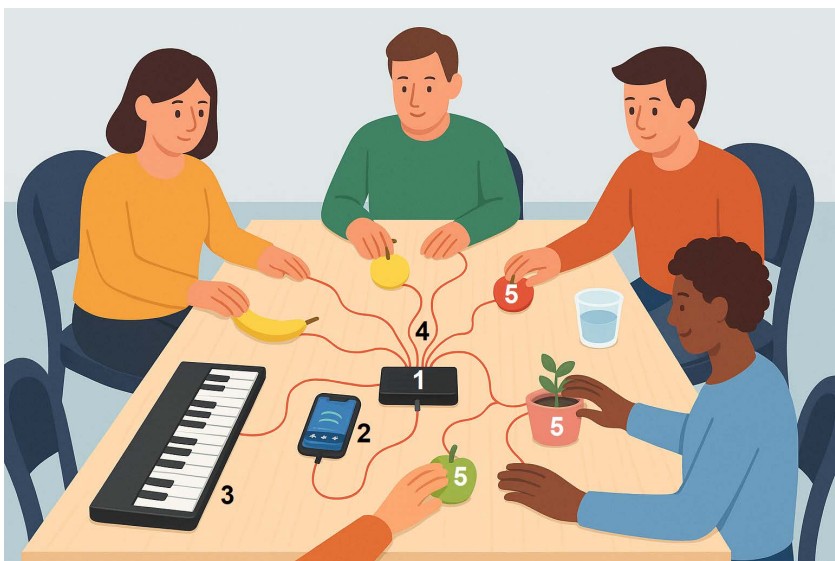

**Fig 2. Setting of interactive MIDI-assisted music therapy sessions.** 1: Playtron MIDI; 2: Cellphone app; 3: Music instrument (keyboard); 4: Copper wire; 5: Conductive objects (fruits, plants, water). Source: Generated with ChatGPT and DALL·E (OpenAI, 2025) [19].

The application of MIDI-enabled devices in music therapy has been documented in first-world countries, but to the best of our knowledge, this is the first documented intervention involving psychiatric patients in which a MIDI controller connected to fruits or other conductive materials was employed to enhance environmental interaction and stimulate a sense of wonder. This innovative approach contributes significantly to the advancement of multisensory music therapy, particularly within the Colombian mental health context. To our knowledge, this is the first documented music therapy intervention in a psychiatric inpatient setting using a MIDI controller connected to conductive materials.

The internal protocol was standardized and remained unchanged throughout the study, with no individual adaptations to the session structure or delivery. Observations of participant engagement and emotional responses did not result in modifications of the intervention.

### Instruments

**Depression, anxiety, and stress scale – 21 (DASS-21).** Evaluates the levels of depression, anxiety, and stress-associated symptoms, with scores categorized according to symptom severity [20,21]. It includes 21 items, divided into three subscales of 7 items each, rate each item on a 4-point Likert scale (0 = Did not apply to me at all to 3 = Applied to me very much or most of the time) [20]. Scores for each subscale are summed and then multiplied by two to match the original DASS-42 scoring system, yielding a final score for each domain; these scores are categorized into severity levels: normal, mild, moderate, severe, and extremely severe [20].

The DASS-21 has demonstrated robust psychometric properties across diverse populations and cultures, including validation in the Colombian population [21,22]. Internal consistency is high, with Cronbach's alpha coefficients reported as 0.94 for the depression subscale, 0.87 for anxiety, and 0.91 for stress, also showing good construct validity and a stable three-factor structure in multiple studies [20].

**Questionnaire of the impact of music therapy sessions in adults (CISMA).** The CISMA is a self-administered pre- and post-session instrument developed by music therapists to evaluate the perceived impact of music therapy on adult patients' well-being [23]. It includes two validated formats: a numerical version, with five visual analog scales ranging from

0 (lowest) to 10 (highest), and a categorical version with five ordinal response options from "not at all" to "completely." The questionnaire assesses five core areas: general well-being, physical pain, relaxation, mood, and perceived social support (companionship). An additional checklist at the end of the post-session form allows patients to indicate specific emotional and social effects perceived during the session. The music therapist selects the appropriate format based on the cognitive and clinical profile of the patient. Completion time ranges from 2 to 4 minutes [23].

The CISMA instrument was originally developed for use in clinical music therapy contexts and emphasizes core dimensions of physical, emotional, and general well-being. Previous clinical and mixed-methods studies have supported its content validity and clinical relevance for capturing patient-reported responses to music therapy interventions, particularly in applied healthcare settings [24]. While further psychometric validation is warranted, available evidence suggests that CISMA items are responsive to short-term changes following music therapy sessions.

Only three items from the CISMA were selected for this study. This selection was intentional and based on their theoretical alignment with outcomes known to be highly sensitive to short-term and session-specific effects of music therapy. Previous research in music therapy consistently indicates that global state well-being, relaxation, or reductions in physiological arousal, and momentary positive affect are among the most immediate and responsive outcomes following a single music therapy session [24,25].

Accordingly, the following CISMA items were selected: "How do you feel now?" as a global indicator of state well-being; "Right now, I feel relaxed" to assess physical tension and physiological calmness; and "Right now, I feel cheerful" as a measure of momentary positive affect. These items correspond to CISMA dimensions most sensitive to within-session change and allowed for the use of a brief assessment format, thereby minimizing participant burden in an acute psychiatric inpatient setting.

**Single item for measuring overall life satisfaction (SWLS).** This measure assesses overall life satisfaction using a single global question, typically phrased as "How satisfied are you with your life overall?", rated on a scale from 0 (completely dissatisfied) to 10 (completely satisfied) [26]. The use of a single-item measure was intended to provide a brief global assessment of life satisfaction while minimizing respondent burden in an acute psychiatric inpatient setting.

Methodological literature suggests that when life satisfaction is conceptualized as a broad and global construct, single-item measures may offer a reasonable approximation for exploratory purposes, particularly in applied or clinical contexts where brevity is essential [26]. However, as single-item measures do not permit the evaluation of internal consistency, factor-analytic properties, or construct validity in the psychometric sense, this measure was used for descriptive and exploratory purposes and interpreted with caution [26].

## Procedure

After providing written informed consent, demographic data were collected and the were randomly assigned to the study groups. Both groups took baseline and post-intervention measurements of the DASS-21, life satisfaction (SWLS), and CISMA questionnaire. Participants in the high-frequency intervention group received five music therapy sessions, with a frequency of one session per day for five consecutive days. Participants in the control group received only one music therapy session during one week of hospitalization (Fig 1).

At the end of the intervention, the patients who belonged to the low intensity group and who continued in the hospitalization process could attend music therapy to complete a total of 5 sessions, thus complying with an equitable treatment of all the participants of the experiment. All participants continued to receive standard inpatient psychiatric care during the study period, with no differences between groups.

## Data analysis

Descriptive analyses were performed to characterize the sample using measures of central tendency and dispersion. Group differences in demographic variables between the low-frequency (1-session) and high-frequency (5-session)

intervention arms were examined using the Mann–Whitney U test for continuous variables and the Chi-square test for categorical variables (i.e., gender). All statistical analyses were conducted using RStudio (version 2024.09.0 + 375). Analyses were conducted on participants who completed the intervention. No imputation methods were applied for missing data, and complete-case analyses were performed.

To evaluate the effect of the intervention on psychological distress and well-being, we conducted repeated measures mixed-effects linear models for each subscale of the DASS-21 (Stress, Anxiety, and Depression) and for the Satisfaction With Life Scale (SWLS). These models accounted for fixed effects (time pre- vs. post-intervention, intervention group, and their interaction), random effects at the participant level, and controlled for relevant covariates, including diagnostic categories. The analysis aimed to capture within-subject variability and address potential confounding effects. Despite initial non-normality in some distributions, model diagnostics supported the use of linear mixed models due to homoscedastic residuals and the absence of influential outliers. Because the three DASS-21 subscales are moderately correlated, they do not constitute independent statistical tests. For this reason, multiplicity adjustments based on independence assumptions (e.g., Bonferroni) are not necessarily optimal for this type of psychometric data. Nonetheless, the potential for Type I error inflation is acknowledged.

Model assumptions were evaluated by inspecting standardized residuals, QQ-plots, residuals-versus-fitted plots, and influence diagnostics; detailed results are provided in **Supplementary Material [S2-S3 Appendices]**.

For the CISMA instrument, a simple linear regression model was used to compare post-intervention scores between groups. Although the Shapiro-Wilk test indicated non-normality of residuals, diagnostic plots supported the assumptions of homoscedasticity and absence of influential outliers, justifying the use of a linear model. Only post-intervention scores of selected CISMA items—focused on emotional change—were analyzed to avoid within-subject pre-post comparisons and reduce potential bias due to spontaneous improvement or regression to the mean.

To explore whether diagnostic differences influenced response to the intervention, we conducted stratified analyses examining the interaction between time (pre- vs. post-intervention) and diagnosis (depression vs. bipolar disorder) on DASS-21 subscale scores. Mixed-effects linear models were fitted separately for each intervention group (experimental vs. control), using depression as the reference category, allowing for random intercepts by participants to account for within-subject variability.

Finally, to further explore potential psychological mechanisms underlying the intervention's impact, a secondary analysis was conducted to examine whether baseline expectations (neutral vs. positive/very positive) moderate changes in DASS-21 scores following music therapy. Specifically, we tested whether having a positive or very positive expectation was associated with greater improvements on DASS-21 subscales, stratified by intervention groups.

### Ethics statement

This study was conducted following the ethical guidelines established in the Declaration of Helsinki of the World Medical Association [27] and the Colombian legislation for research involving human participants [28].

Both the research protocol and the informed consent were approved by an independent research ethics committee (CEI Campo Abierto Ltda, approval act No. 209 of July 2024). All study participants signed the informed consent document before any research procedure was performed. The capacity to provide consent was determined by the treating psychiatrist and documented in the patient's clinical records. In addition, to guarantee the principle of equity of the participants assigned to the control group and considering that they received fewer music therapy sessions than the high-frequency group, once the measurement of study outcomes was completed, they were allowed access to the opposite intensity of sessions.

To ensure nonmaleficence, the investigators who applied for the tests considered immediately suspending the study when they noticed any risk or harm to the health of the participants or discontinuing the study on an individual basis when they noticed a particular risk in a research participant.

# Results

A total of 106 patients were screened, of whom 91 participants were enrolled and randomly assigned. Fig 1 describes the recruitment and follow-up of study participants. For the primary outcome analysis, 37 participants in the intervention group and 37 in the control group were included. Table 1 presents the demographic and clinical characteristics of the sample. The diagnostic distribution in our sample reflects the typical clinical profile of psychiatric inpatients in the study setting, which is predominantly female, and with primary diagnoses of depressive disorders, followed by bipolar and psychotic disorders [18]. As can be seen, no significant baseline differences were found between the assignment groups regarding gender, schooling, diagnostic axis, and expectation of music therapy (Table 1).

The mixed-effects model revealed a significant interaction between time and intervention group for the DASS-21 stress subscale. Both groups showed post-intervention improvements, but the high-frequency group exhibited a greater stress reduction (−6.19 points) than the low-frequency group, indicating that higher session intensity was associated with greater stress relief (p = 0.023) (Table 2) (Fig 3).

For the anxiety subscale, the high-frequency group showed a significant decrease of 5.41 points from pre- to post-intervention (p < 0.001). Although the low-frequency group also improved, and the interaction effect (group × time) was not statistically significant between groups (difference = −1.19 points, p = 0.339) (Table 2).

Regarding the depression subscale, both groups significantly improved post-intervention, although the difference between groups did not reach statistical significance (p = 0.270).

For the Satisfaction With Life Scale (SWLS), the high-frequency group experienced a significant post-intervention increase of 11.6 points (p = 0.012). However, no statistically significant difference in change scores was observed between groups.

For CISMA Item 1, participants in the high-frequency group reported marginally higher scores (+0.71 points) than those in the low-frequency group; however, this difference did not reach statistical significance (p = 0.184). For Item 3, mean scores were slightly lower in the high-frequency group (–0.15 points), with no significant difference between groups

**Table 1. Demographic and clinical characteristics.**

| | | High-frequency Intervention Group (n = 37) | Low-frequency Control Group (n = 37) | p value |
|---|---|---|---|---|
| | | n (%) | n (%) | |
| Age | Mean (S.D) | 37.5 (±14.9) | 34.9 (±16.4) | 0.31* |
| Gender | Male | 13 (35.1) | 6 (16.2) | 0.11** |
| | Female | 24 (64.9) | 31 (83.8) | |
| Level of education | Postgraduate | 15 (40.5) | 10 (27) | 0.38* |
| | Professional | 5 (13.5) | 9 (24.3) | |
| | Bachelor's, Technological | 11 (29.7) | 11 (29.7) | |
| | Less than Bachelor's | 6 (16.2) | 7 (18.9) | |
| Expectations of music therapy | Very Positive | 21 (56.8) | 25 (67.6) | 0.27* |
| | Positive | 11 (29.7) | 10 (27) | |
| | Neutral | 5 (13.5) | 2 (5.4) | |
| Diagnostic group | Depressive Disorder | 23 (62.2) | 22 (59.5) | 0.99* |
| | Bipolar Disorder | 7 (18.9) | 9 (24.3) | |
| | Anxiety/Stress Disorder | 2 (5.4) | 4 (10.8) | |
| | Psychosis | 4 (10.8) | 1 (2.7) | |
| | Other | 1 (2.7) | 1 (2.7) | |

*Mann-Whitney test. ** Chi-squared test.

**Table 2. Repeated measures mixed model analysis of DASS-21 and SWLS scores by intervention group and time.**

| Scale | Timepoint | High-frequency group (n = 37) Mean ± SD | Low-frequency group (n = 37) Mean ± SD | Group (Baseline group comparison) * p-value | Time (Pre-Post comparison) * p-value | Group×Time * p-value |
|---|---|---|---|---|---|---|
| DASS-21 Stress | Pre | 12.5 ± 6.77 | 11.6 ± 6.69 | −0.24 p = 0.809 | −6.96 p < 0.001 | 2.32 **p = 0.023** |
| | Post | 6.49 ± 5.55 | 7.03 ± 5.81 | | | |
| DASS-21 Anxiety | Pre | 11.6 ± 5.11 | 11.5 ± 5.53 | −0.11 p = 0.912 | −6.184 p < 0.001 | 0.96 p = 0.339 |
| | Post | 6.24 ± 5.01 | 7.30 ± 5.23 | | | |
| DASS-21 Depression | Pre | 12.7 ± 4.94 | 12.4 ± 5.74 | −0.62 p = 0.539 | −6.58 p < 0.001 | 1.11 p = 0.270 |
| | Post | 6.49 ± 4.38 | 9.11 ± 5.87 | | | |
| SWLS | Pre | 54.7 ± 32.2 | 60.4 ± 33.2 | 0.82 p = 0.412 | 2.58 p < 0.05 | −0.16 p = 0.875 |
| | Post | 66.3 ± 28.6 | 71.0 ± 22.8 | | | |

*t value (Mixed Model Analysis).

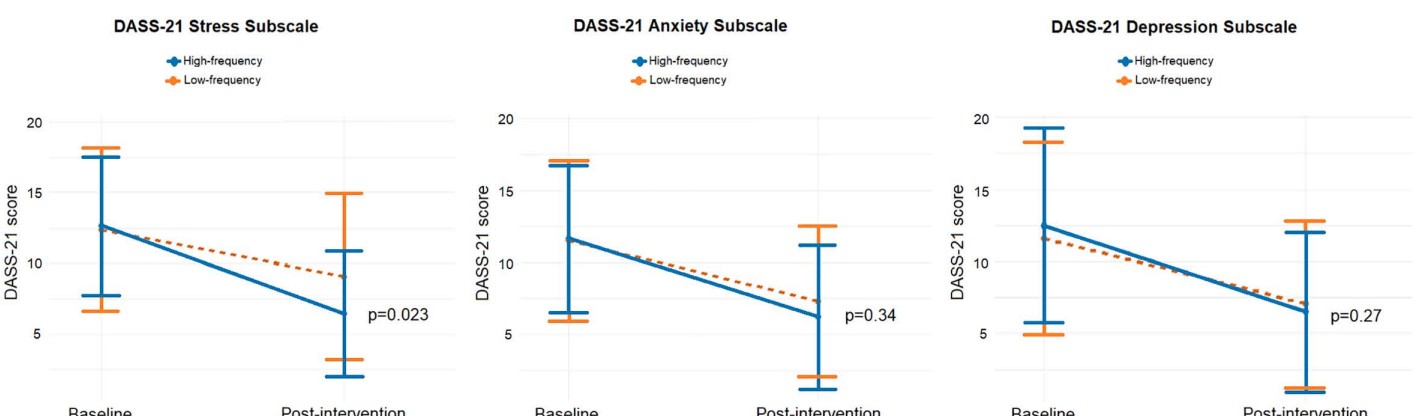

**Fig 3. Comparison of DASS-21 between groups.** The difference in response to depression, anxiety, and stress in both groups. Interaction effect from the linear mixed model.

(p = 0.787). Similarly, for Item 4, the high-frequency group showed a modest average increase (+0.53 points), though again without statistical significance (p = 0.346). Across all models, explained variance was minimal, suggesting limited sensitivity of these items to detect between-group differences in perceived emotional change (Table 3).

In the experimental group, significant differential effects by diagnosis were observed in the depression subscale. Participants with bipolar disorder showed significantly lower depression scores compared to those with depression (β = −2.96, p < 0.001). Moreover, the pre-post change was significantly greater for the bipolar group (β = 2.26, p = 0.005) (Table 4).

For the anxiety subscale, a greater reduction was also observed in the bipolar group compared to the depression group (β = −1.85, p = 0.011), though the between-diagnosis difference was not statistically significant (β = 1.09, p = 0.166). A similar, non-significant trend was observed for stress, with lower scores and greater reductions in the bipolar group (β = −1.27, p = 0.061; change β = 0.78, p = 0.303) (Table 4).

For the stress and anxiety subscales, interaction effects between time and expectation were not statistically significant in either group (Table 5).

In the case of **depression**, participants in the high-frequency group with very positive expectations compared to those with neutral expectations had greater symptom improvement in psychological distress (β = −5.92, p = 0.037) (Table 5).

**Table 3. Comparison of post-intervention CISMA item scores using a linear model.**

| Post – CISMA | High-frequency group (Post final session) n = 36 Mean (SD) | Low-frequency group (Post final session) n = 37 Mean (SD) | β (95% CI) | Standard Error | t, p-value |
|---|---|---|---|---|---|
| Item 1 | 8.08 (2.02) | 7.38 (2.44) | 0.71 | 0.53 | 1.34, p = 0.18 |
| Item 3 | 7.31 (2.30) | 7.46 (2.56) | −0.15 | 0.57 | −0.27, p = 0.79 |
| Item 4 | 7.69 (1.97) | 7.16 (2.75) | 0.53 | 0.56 | 0.94, p = 0.35 |

CISMA Visual Analog Scale (0–10): Item 1: "How do you feel now", Item 3: "Right now, I feel relaxed". Item 4: "Right now, I feel cheerful". One participant from the high-frequency group was lost to follow-up.

**Table 4. Interaction between diagnosis (depression vs bipolar) and time (pre-post) on DASS-21 scores by intervention group (linear mixed model).**

| DASS-21 Subscale | Group | Bipolar vs Depression β (coef.) | Standard Error, p value | Pre-Post Change × Diagnosis β (coef.) | Standard Error, p value |
|---|---|---|---|---|---|
| Stress | High-frequency | −1.27 | 0.67, p = 0.061 | 0.78 | 0.75, p = 0.303 |
| | Low-frequency | −0.21 | 1.00, p = 0.838 | 0.68 | 0.96, p = 0.484 |
| Anxiety | High-frequency | −1.85 | 0.71, p = 0.011 | 1.09 | 0.77, p = 0.166 |
| | Low-frequency | −0.42 | 0.93, p = 0.655 | 0.49 | 0.89, p = 0.581 |
| Depression | High-frequency | −2.96 | 0.84, p < 0.001 | 2.26 | 0.75, **p = 0.005** |
| | Low-frequency | −1.54 | 1.07, p = 0.153 | 1.49 | 0.89, p = 0.101 |

**Diagnosis groups:** depression (n = 45), bipolar disorder (n = 16). Negative coefficients indicate greater post-intervention reductions in DASS-21 subscale scores compared to the depression group.

**Table 5. Moderating effect of participant expectations on changes in DASS-21 scores by intervention group.**

| DASS-21 Subscale | Interaction Neutral vs other expectations | High-frequency β (coef.) | Standard error, p-value | Low-frequency β (coef.) | Standard error, p-value |
|---|---|---|---|---|---|
| Stress | vs Positive expectation | −2.73 | 2.76, p = 0.330 | −3.30 | 4.22, p = 0.439 |
| | vs Very positive expectation | −4.19 | 2.55, p = 0.109 | −6.48 | 4.00, p = 0.115 |
| Anxiety | vs Positive expectation | 0.51 | 3.00, p = 0.867 | 1.90 | 4.02, p = 0.639 |
| | vs Very positive expectation | 0.08 | 2.77, p = 0.978 | −1.08 | 3.81, p = 0.779 |
| Depression | vs Positive expectation | −3.49 | 2.96, p = 0.246 | −1.70 | 4.26, p = 0.692 |
| | vs Very positive expectation | −5.92 | 2.73, **p = 0.037** | −0.90 | 4.04, p = 0.825 |

**Expectation levels:** neutral (n = 7), positive (n = 21), very positive (n = 46). Negative coefficients indicate greater post-intervention reduction in DASS-21 subscale scores compared to the neutral expectation group.

## Discussion

To the best of our knowledge, this is the first randomized controlled trial to evaluate the effects of high-frequency compared with low-frequency MIDI-assisted group music therapy delivered over a short intervention period during psychiatric hospitalization. The results indicate that participants receiving higher-frequency music therapy experienced a greater reduction in stress symptoms compared with those receiving a single session. No statistically significant between-group differences were observed for depressive or anxiety symptoms, global life satisfaction, or subjective perceptions of the

music therapy experience. A subgroup analysis further suggested greater reductions in depressive symptoms among patients with bipolar disorder receiving high-frequency intervention.

Although both groups demonstrated significant within-group improvements across all measured outcomes, the absence of a non-intervention control group limits causal attribution, as all participants received standard multimodal inpatient psychiatric care, including psychopharmacological treatment. Consequently, symptom improvements cannot be attributed exclusively to music therapy. Nevertheless, the randomized design, the use of linear mixed-effects models accounting for repeated measures, and the consistency of improvement across validated instruments (DASS-21, SWLS, CISMA) suggest that music therapy may have contributed to short-term symptom reduction, particularly in the stress domain.

Current evidence supports the integration of music therapy as an adjunctive intervention in acute psychiatric settings, with a capacity to reduce overall psychiatric symptomatology, particularly affective symptoms such as depression, anxiety, and stress [15,17]. While meta-analyses in specific diagnostic groups, including post-traumatic stress disorder, suggest a dose–response relationship in which longer treatment duration is associated with greater clinical benefit [16], research focused on acute hospitalization and depressive disorders indicates that short- to medium-duration interventions can produce clinically relevant short-term effects [17]. Moreover, controlled studies conducted during acute psychotic episodes have reported improvements in depressive and negative symptoms, as well as reductions in length of hospital stay, suggesting a rapid process of emotional stabilization [15].

Beyond long-term functional recovery, recent literature emphasizes the value of music therapy in facilitating immediate emotional regulation, emotional expression, and the development of short-term coping strategies in highly complex clinical contexts [29]. In particular, systematic reviews have shown that brief or single-session music-based interventions can enhance motivation and treatment readiness in detoxification units and acute psychiatric wards, thereby acting as a catalyst for therapeutic engagement [15]. Taken together, these findings indicate that music-based interventions may offer clinically meaningful benefits for the management of acute affective symptoms and patient engagement, although evidence regarding the long-term sustainability of these effects in the absence of continued intervention remains limited [30].

Consistent with this literature, both intervention groups in the present study showed reductions in anxiety and depressive symptoms following treatment. Similar calming and anxiolytic effects of music-based interventions have been reported in medical contexts, including reductions in pain and anxiety during medical procedures [10]. Systematic reviews summarized by Tang et al. and Zhao et al. further suggest that the duration and volume of musical interventions may influence emotional outcomes [10,31].

Regarding intervention intensity, both groups demonstrated clinical improvement; however, participants receiving higher-frequency music therapy exhibited a statistically significant reduction in stress levels compared with those receiving a single session. This finding aligns with emerging evidence suggesting that cumulative exposure and session frequency may influence treatment effects, although the dose–response relationship appears to be diagnosis-specific and context-dependent. In depressive disorders, short- to medium-duration interventions (approximately 1–12 sessions) have been associated with clinically relevant effects, supporting the feasibility of brief but intensive interventions during acute phases of care [17].

Additional studies support the feasibility of short-term or intensive music-based interventions across clinical populations. Functional improvements have been reported following one to seven weekly sessions in patients with Parkinson's disease [6], and the SYNCHRONY trial highlighted the potential value of higher-frequency music therapy in individuals with long-term depression [32]. Although the intervention period in the present study was necessarily brief, the observed stress-related differences suggest that even short-term intensive formats may yield measurable benefits in acute care settings.

Nevertheless, there remains limited evidence defining the minimum intensity required to generate clinically significant effects. For example, Colin et al. reported reductions in stress following daily 30-minute music listening over three weeks

in geriatric healthcare workers [33]. Although this intervention differed from structured music therapy, the similarity in session duration and stress-related outcomes is noteworthy.

Similarly, Zhou et al. demonstrated that the combination of music interventions with Progressive Muscle Relaxation was effective in reducing anxiety and depression and improving quality of life in oncology patients [34], with associated reductions in cortisol levels. This intervention structure resembles the approach used in the present study and provides biological plausibility for the observed stress-related effects, although outcomes in the present trial were assessed using psychological rather than physiological measures.

Proposed mechanisms suggest that music may modulate stress and anxiety through neuroendocrine and autonomic pathways, including effects on the peripheral nervous system and neurotransmitter systems involved in mood regulation [35]. These mechanisms may partly explain the greater stress reduction observed in the high-frequency group, consistent with findings from studies combining music interventions and relaxation techniques [34,35].

Stress-reducing effects of music-based interventions have also been reported in diverse populations, including operating room staff [36], students facing examinations [37], patients with osteosarcoma [38], and obstetric populations [9]. While these populations differ from psychiatric inpatients, the consistency of stress-related findings across settings provides indirect support for the stress-domain results observed in this study.

Interpretation of the stress-domain interaction requires caution due to multiple comparisons. Although the DASS-21 subscales are correlated components of a multidimensional instrument, separate testing increases the risk of type I error. Classical Bonferroni correction may be overly conservative in this context; however, if applied, the observed interaction would not meet the adjusted significance threshold. Accordingly, this finding should be interpreted as exploratory and considered alongside the consistent within-group improvements across domains. In addition, the use of a limited subset of CISMA items, while appropriate for assessing immediate session-related changes in an acute inpatient context, restricts conclusions regarding broader emotional or functional domains.

Consistent with previous literature, no significant between-group differences were observed for global life satisfaction, paralleling findings reported by Lee et al. [6], which noted limited effects of music therapy on quality of life despite improvements in affective symptoms. The use of a single-item life satisfaction measure may also have reduced sensitivity to detect subtle changes [26].

Additional studies have reported reductions in anxiety and depressive symptoms across medical populations [39,40], as well as physiological stress responses following brief music interventions in critically ill patients [41,42]. While these findings support the broader stress-modulating potential of music-based interventions, further research is needed to clarify underlying mechanisms and long-term effects.

Finally, this study explored the role of patient expectations. Although moderation analyses did not reveal significant interaction effects for most participants, individuals with highly positive expectations showed greater reductions in depressive symptoms, consistent with previous findings highlighting the influence of expectancy on treatment engagement and outcomes [31]. These results suggest that patient expectations may modulate response in some individuals and should be considered in future research.

## Limitations

This study presents several limitations that should be considered when interpreting the results. First, it was conducted in a single urban psychiatric referral center, which may restrict the generalizability of the findings to other clinical contexts such as outpatient care, general hospitals, or rural settings. Second, the nature of the intervention precluded participant blinding, potentially introducing performance bias, particularly relevant given the observed association between therapeutic outcomes and participants' expectations. Another limitation relates to the discrepancy between the a priori sample size estimation and the statistical model used for the primary analysis. The original calculation was based on a two-sample t test, whereas the final analysis required a repeated-measures structure using linear mixed models. Because power

estimation for mixed models is substantially more complex, the achieved sample size may offer reduced power to detect interaction effects or smaller-than-expected differences between groups.

A major limitation of this study is that the trial registration was conducted post hoc. In Colombia, prospective registration of clinical trials on international platforms is not a mandatory requirement for conducting research; however, the study was retrospectively registered in order to meet the submission requirements of the journal. The study, therefore does not have a publicly registered protocol; instead, it was conducted under a complete written protocol developed prior to data collection and approved by an Institutional Ethics Committee in accordance with national regulations (CEI Campo Abierto Ltda, approval act No. 209 of July 2024), and all procedures adhered to national and international ethical standards. Nevertheless, the retrospective registration remains a methodological limitation, as it does not allow for public verification of the study timeline before its initiation.

The study focused exclusively on inpatients receiving short-term music therapy, limiting its applicability to populations undergoing longer-term treatments. Additionally, the sample was predominantly composed of individuals with depressive and bipolar disorders, which may constrain extrapolation to other psychiatric diagnoses.

Measurement-related limitations also emerged. While validated instruments like the DASS-21 and the Satisfaction With Life Scale were employed, the CISMA scale remains under development and may lack psychometric robustness. Furthermore, the SWLS and CISMA did not appear sufficiently sensitive to detect specific changes attributed to music therapy. Complementing self-report measures with objective assessments in future research is recommended.

Finally, data on pharmacological treatment, including medication types and dosages, were not systematically recorded or compared across groups. Although all participants were under psychiatric medication, unmeasured pharmacological differences may have influenced outcomes and constitute a potential confounding factor.

Given the impossibility of blinding participants and therapists in music therapy interventions, future research could benefit from incorporating objective clinical indicators such as length of stay, psychiatric treatment response, or medication adjustments as secondary outcomes. These measures would help reduce the risk of bias associated with subjective assessments and offer a more comprehensive evaluation of treatment effects.

## Adverse events

Participants were systematically monitored for adverse events throughout the study. The following categories were assessed: psychological destabilization (e.g., increased anxiety, emotional agitation, dissociative reactions), behavioral disturbances (e.g., aggression, withdrawal), and somatic complaints potentially related to overstimulation. Monitoring was conducted through direct observation by facilitators, communication with clinical staff, and self-reports from participants.

Adverse events were actively monitored throughout the intervention period through clinical observation during music therapy sessions and participant self-report. Any physical or psychological discomfort potentially related to the intervention was documented and reviewed by the clinical team. No adverse events related to the intervention were reported in either study group during the intervention period. This favorable safety profile may be attributed to careful clinical screening of participants, delivery of sessions by an experienced music therapist, continuous on-site supervision by a psychiatry resident and a physician trained in psychological first aid, and the use of a structured, supportive group setting. Notably, no participants discontinued the intervention due to distress, and qualitative feedback indicated high levels of emotional safety and engagement.

## Conclusion

This study provides preliminary evidence supporting the effectiveness of brief, structured group music therapy, demonstrating significant improvements across both intervention groups in stress, anxiety, depression, and life satisfaction in hospitalized psychiatric inpatients, with greater benefit in the high-frequency group for stress reduction.

Future studies with larger samples are needed to confirm these findings and further explore the interplay between treatment intensity, diagnostic subgroups, and psychological expectations.

 

## Supporting information

**S1 Appendix. TIDieR checklist.**
(DOCX)

**S2 Appendix. Model diagnostics and additional statistical outputs.**
(DOCX)

**S3 Appendix. Full intervention protocol.**
(PDF)

**S1 File. CONSORT_2025_checklist_(revised)_9Feb2026.**
(DOCX)

## Acknowledgments

Primarily to our patients, who gave us their time and the opportunity to contribute to the growth of knowledge. To our entire team, comprising our therapists, medical support staff, researchers, and consultants. As well as to the Clinica Montserrat – Hospital Universitario, for offering its support and facilities.

## Author contributions

**Conceptualization:** Manuel Esteban-Cárdenas, Carlos Torres-Delgado, Adrián Hidalgo-Valbuena, Eugenio Ferro.

**Data curation:** Manuel Esteban-Cárdenas.

**Formal analysis:** Carlos Torres-Delgado, Eugenio Ferro.

**Funding acquisition:** Manuel Esteban-Cárdenas, Eugenio Ferro.

**Investigation:** Manuel Esteban-Cárdenas, Ana Gómez-Puentes, Carlos Torres-Delgado, Adrián Hidalgo-Valbuena, Eugenio Ferro.

**Methodology:** Manuel Esteban-Cárdenas, Eugenio Ferro.

**Project administration:** Manuel Esteban-Cárdenas, Ana Gómez-Puentes.

**Resources:** Adrián Hidalgo-Valbuena.

**Software:** Manuel Esteban-Cárdenas, Carlos Torres-Delgado, Eugenio Ferro.

**Supervision:** Eugenio Ferro.

**Validation:** Ana Gómez-Puentes, Carlos Torres-Delgado, Eugenio Ferro.

**Visualization:** Ana Gómez-Puentes.

**Writing – original draft:** Manuel Esteban-Cárdenas, Ana Gómez-Puentes, Carlos Torres-Delgado, Adrián Hidalgo-Valbuena, Eugenio Ferro.

**Writing – review & editing:** Manuel Esteban-Cárdenas, Ana Gómez-Puentes, Carlos Torres-Delgado, Adrián Hidalgo-Valbuena, Eugenio Ferro.

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
