## [Decision Letter · Decision Letter 0]

9 Apr 2025

Dear Dr. Ferro,

Thank you for submitting your manuscript to PLOS ONE. After careful consideration, we feel that it has merit but does not fully meet PLOS ONE’s publication criteria as it currently stands. Therefore, we invite you to submit a revised version of the manuscript that addresses the points raised during the review process.

.

We look forward to receiving your revised manuscript.

Kind regards,

Sascha Köpke

Academic Editor

PLOS ONE

Reviewers' comments:

Reviewer's Responses to Questions

**Comments to the Author**

1. Is the manuscript technically sound, and do the data support the conclusions?

Reviewer #1: Partly

Reviewer #2: Partly

2. Has the statistical analysis been performed appropriately and rigorously?

Reviewer #1: No

Reviewer #2: No

3. Have the authors made all data underlying the findings in their manuscript fully available?

Reviewer #1: Yes

Reviewer #2: Yes

4. Is the manuscript presented in an intelligible fashion and written in standard English?

Reviewer #1: Yes

Reviewer #2: Yes

Reviewer #1: This study presents findings from a randomized controlled trial comparing the effectiveness of high-frequency versus low-frequency music therapy. My evaluation focuses on the study's methodological aspects.

First, I question whether the sample size calculation was conducted correctly. The authors state that they aimed to detect a mean difference of 4 between the experimental and control groups, with an estimated standard deviation of 5. However, it is unclear what this difference of 4 represents. Typically, sample size calculations are based on effect size, which must be explicitly linked to the intervention outcome. Additionally, the software and statistical model used to determine this number should be specified.

One of the inclusion criteria is “pharmacological treatment with medication adjustments.” Given that different medications may be prescribed based on condition severity, they could also influence the effectiveness of the music therapy intervention. Have the authors collected and compared medication data between groups?

The authors conducted a preliminary Shapiro-Wilk test for normality, which yielded significant results. Consequently, they opted for non-parametric tests throughout the analysis. However, this choice precludes the use of more robust methods, such as repeated measures analysis. The significant results in Table 2 indicate only a difference in DASS-21 scores between groups. Without a formal test that incorporates both pre- and post-intervention data, it remains unclear whether the effect is greater in the intervention group. Statistical tests of normality based simply on p-values are often overly stringent, the authors may consider visual examination as well. In situations where the deviation from normality is not substantial, conventional approach can still be used.

Additionally, multiple outcomes were analyzed without adjustments for multiple comparison bias. The only significant p-value (0.03) is relatively weak and would likely lose significance with even a mild adjustment. Thus, the evidence supporting the effectiveness of high-frequency music therapy is not strong.

The authors also conducted a series of correlations between expectation levels and various outcomes. However, since expectation levels are not directly related to the intervention itself, this analysis falls outside the study’s scope and does not support the intervention’s effectiveness.

Several wording issues should also be addressed:

1. The term “bilateral” significance level is not standard.

2. In the data analysis section, the authors describe a “descriptive analysis” that characterizes the sample, including measures of central tendency and dispersion (percentiles). However, dispersion should be measured by variance/standard deviation when using the mean and by interquartile range when using the median. These essential dispersion measures are missing from the results.

Reviewer #2: The authors present a novel randomised controlled trial to test for differences between high and low intensity group music therapy on psychiatric inpatient wards. The study appears to have been generally well conducted although the manuscript would benefit from greater clarity and detail in a number of areas. Greater attention to how recruitment and group delivery were managed would be of interest given the successful retention in the trial and the challenges of high patient turnover. More detail regarding session attendance is also important given the focus of intensity in this trial. The manuscript at present is limited by its lack of music therapy literature pertaining to work in acute hospitals and psychiatry (both introduction and discussion); lack of detail regarding study procedures and lack of detailing pertaining to the intervention. Following CONSORT guidance on reporting randomised controlled trials and TiDier template for reporting interventions will help. Future revisions should refer explicitly to these guidelines/checklists:

https://www.equator-network.org/reporting-guidelines/consort/

http://www.tidierguide.org/

Some further detail is below to assist the authors with major revisions:

1. Literature review

Key literature on music therapy within inpatient settings is missing, as is the unique inpatient context. The authors are encouraged to perform a search on pubmed or similar with terms "music therapy" "mental health" "inpatient" to summarise knowledge to date. See particularly work by Michael J. Silverman (where he has offered many different high intensity interventions), Catherine Carr (see systematic review of music therapy for acute adult psychiatric inpatients in PLOS ONE for an overview of practice and methods up to 2012 - https://pmc.ncbi.nlm.nih.gov/articles/PMC3732280/ and PhD thesis) and wider considerations of inpatient music therapy practice.

2. Procedures

Timing of the outcome assessments is not clear. More generally the procedure from recruitment into the study, randomisation, intervention delivery and followup needs greater clarity. The protocol published under ISRCTN provides an interesting chart suggesting patients were recruited in daily cohorts. Some explanation of how many groups ran (was it one each of high intensity and low intensity or several over the course of the study?) and at what point assessments were taken would be helpful. Please define the time window allowed to collect followup data.

Please provide a brief description of how each of the scales is rated and scored along with psychometric properties.

Please provide more detail on your secondary analysis of patient expectations and the methods used to analyse this.

3. Intervention

Please use the TiDier template to provide key information about the group music therapy offered in this trial.

The intervention described sounds quite unusual in respect to music therapy practice on inpatient wards. To what degree is this approach used in other hospitals? If it is the first time such an approach has been used, this should be made explicit as it limits generalisability. I would also suggest rephrasing of the intervention as group music therapy using MIDI controlled instruments in the title and throughout to make clear that this is one group approach among many.

Were multiple participants using a PLAYTRON simultaneously, or was it one controller that was played by one person and others listened? It would help to make clear the extent to which group members were able to jointly create music within sessions.

What certification/training does your music therapist hold and how does it relate to professional standards nationally and internationally? This should be stated in the report.

Where were the groups held? On or off the ward/ in an open or closed space.

Given the focus on the difference in intensity it is important to get a sense of how many sessions were available to participants (was it only 5 / 1 session, or could participants attend subsequent weeks ie. if they attended for 2 weeks, they could access up to 10/2 sessions) and the number of sessions actually attended by participants in both arms - please summarise attendance and group size (mean, s.d., range).

To what extent did group membership change while the groups were running? A flow of participants would also help to demonstrate the extent group membership was stable or not.

4. Adverse events

It's unusual to have no adverse events in the course of a study. Can you state briefly what sorts of events you assessed for and what might have helped to ensure none occurred during the study.

5. Sample diversity

The discussion would benefit from further detail regarding the sample. Could you explain why numbers of people with psychosis were much lower than depression or anxiety? Similarly it is worth commenting that the study included more women than men. Please also see comment below regarding sampling strategy/length of hospital stay from ISRCTN protocol. It would be helpful to present average length of hospital stay at point of recruitment along with how many were lost during intervention due to discharge.

6. Results

There is at least one inaccurate percentage reported in table 1 - suggest these figures are checked.

Why are median scores presented in table 2? Mean and s.d. would help to give a better sense of spread of scores.

It would be helpful to explain the two assessment points earlier on in the text and how data from each of these points were used.

7. Patient expectancy:

Regarding patient expectations of music therapy - did you consider doing a secondary analysis of outcomes, controlling for expectancy rather than a single regression with your overall sample? Do the outcomes change at all?

Lines 312-4 "Additionally, in this study, we found the influence of a suggestive effect of music therapy, since the participants who reported greater expectations before music therapy sessions had better outcomes in depressive symptoms, stress-associated, global life satisfaction, and perception of the impact of music therapy."

The association between patient expectations and outcome is hard to disentangle as patients would not have signed up for the study without some positive expectation of attending so the degree to which expectation influences outcome is always going to be limited by a ceiling effect. Both arms have similar proportions of high/medium/neutral expectation so we can say that the differing frequency appears to have an effect but that expectations might inflate this. A different study would be required to test the specific expectancy effects but you could control for expectancy in your results and report whether the effects you initially report hold true (but with the limitations that this is a secondary/post-hoc analysis).

Lines 279-280:"Positive correlations were found, employing a matrix, between the expectation of music therapy and final life satisfaction (P<0.001), CISMA final total score in the first (P=0.005) and last measurement (P=0.001)" - I did not understand this. Please explain/make clearer.

Discussion:

As per literature review/introduction, this needs re-writing with a stronger focus on comparison with existing music therapy/mental health papers rather than wider studies.

Line 316-7 "When evaluating the intensity of sessions needed to find an effect, the meta-analysis of Aalbers et al. found that adding twelve music therapy sessions to the usual treatment is more effective in reducing symptoms of depression, anxiety, and improved functioning (21)" - I could not find this information in the referred paper. Please check the citation. You might also want to refer to Gold et al's paper on dose response: https://pubmed.ncbi.nlm.nih.gov/19269725/

8. Comparison with published protocol on ISRCTN

"All ppt will participate in all intervention modes high and low intensity" – did high intensity also have low intensity offered afterwards or did your protocol mean just that low intensity participants were offered high intensity afterwards? How did this fit with assessments, and did this mean that additional participants (not being assessed) were in later groups?

"Priority will initially be given to patients with longer hospitalization durations to maximize intervention exposure. As recruitment progresses and patients with extended stays are enrolled, focus will gradually shift toward those with shorter hospitalizations. This approach is designed to maintain consistent group sizes for the intervention sessions, facilitating robust and enriched group therapy sessions throughout the recruitment period."

Please ensure this detail is in the procedures section of the paper - and also acknowledged as a limitation.

**Do you want your identity to be public for this peer review?** For information about this choice, including consent withdrawal, please see our For information about this choice, including consent withdrawal, please see our Privacy Policy .

Reviewer #1: No

Reviewer #2: **Yes:** Dr Catherine CarrDr Catherine Carr

While revising your submission, please upload your figure files to the Preflight Analysis and Conversion Engine (PACE) digital diagnostic tool, https://pacev2.apexcovantage.com/ . PACE helps ensure that figures meet PLOS requirements. To use PACE, you must first register as a user. Registration is free. Then, login and navigate to the UPLOAD tab, where you will find detailed instructions on how to use the tool. If you encounter any issues or have any questions when using PACE, please email PLOS at . PACE helps ensure that figures meet PLOS requirements. To use PACE, you must first register as a user. Registration is free. Then, login and navigate to the UPLOAD tab, where you will find detailed instructions on how to use the tool. If you encounter any issues or have any questions when using PACE, please email PLOS at figures@plos.org . Please note that Supporting Information files do not need this step.. Please note that Supporting Information files do not need this step.

---

## [Author Response · Author response to Decision Letter 1]

9 Jun 2025

Reviewer 1 – Comment 1:

First, I question whether the sample size calculation was conducted correctly. The authors state that they aimed to detect a mean difference of 4 between the experimental and control groups, with an estimated standard deviation of 5. However, it is unclear what this difference of 4 represents. Typically, sample size calculations are based on effect size, which must be explicitly linked to the intervention outcome.

Response:

We appreciate the reviewer’s insightful observation. We have revised the manuscript to clarify the rationale behind our sample size calculation. The originally reported difference of 4 points corresponds to an expected mean difference in outcome scores based on prior literature. However, for clarity and precision, we now express this as an effect size of 0.6, following the findings reported by Zhang et al. (2022), which served as a reference for expected clinical impact in music therapy interventions. Accordingly, the sample size was calculated using a two-tailed test with α = 0.05 and power = 0.80, assuming a 1:1 allocation ratio and a medium effect size (Cohen’s d = 0.6). This yielded a required sample of 30 participants per group. Accounting for an anticipated 20% dropout rate, we aimed to recruit a total of 74 participants (37 per group). The revised text is now included in the Methods section.

Reviewer 1 – Comment 2:

Additionally, the software and statistical model used to determine this number should be specified.

Response:

Thank you for this observation. We have now included the software version used for the sample size calculation in the Methods section. Specifically, the calculation was performed using RStudio (version 2024.09.0+375). The statistical model applied was based on a two-sample t-test assuming equal variances and a two-sided significance level of 0.05.

Reviewer 1 – Comment 3:

One of the inclusion criteria is “pharmacological treatment with medication adjustments.” Given that different medications may be prescribed based on condition severity, they could also influence the effectiveness of the music therapy intervention. Have the authors collected and compared medication data between groups?

Response:

We thank the reviewer for this important observation. We acknowledge that different pharmacological treatments could have influenced participants’ responses to the music therapy intervention. However, medication type and dosage were not systematically collected or compared between groups. This has now been explicitly recognized as a limitation in the revised manuscript. Additionally, we have added a new subsection on study limitations within the Discussion section to address this and other methodological considerations identified during the revision process.

Reviewer 1 – Comment 4:

The authors conducted a preliminary Shapiro-Wilk test for normality, which yielded significant results. Consequently, they opted for non-parametric tests throughout the analysis. However, this choice precludes the use of more robust methods, such as repeated measures analysis. [...] In situations where the deviation from normality is not substantial, a conventional approach can still be used.

Reviewer 1 – Comment 5:

Additionally, multiple outcomes were analyzed without adjustments for multiple comparison bias. The only significant p-value (0.03) is relatively weak and would likely lose significance with even a mild adjustment. Thus, the evidence supporting the effectiveness of high-frequency music therapy is not strong.

Comment 4 & 5 Combined Response:

Thank you for these important observations. In response, we reanalyzed the data using mixed-effects linear models to account for both fixed effects (time, group, and their interaction) and random effects, while controlling for relevant covariates including patients' diagnostic categories and time. This modeling approach provides greater robustness by accommodating within-subject variability and helps reduce the risk of inflated Type I error due to multiple comparisons.

Although initial Shapiro-Wilk tests indicated non-normality, diagnostic plots showed homoscedastic residuals and no influential outliers, supporting the validity of these models.

Regarding the DASS-21, both groups improved post-intervention. The stress subscale showed a significant group-by-time interaction (p = 0.023), with the five-session group exhibiting a greater reduction (−6.19 points), suggesting a frequency-dependent effect.

In the anxiety subscale, the five-session group had a significant pre-post reduction (−5.41 points, p < 0.001), but the group-by-time interaction was not significant (p = 0.339), indicating similar improvements across groups. Depression scores also improved significantly in both groups, though no significant group-by-time difference was found (p = 0.27). For life satisfaction (SWLS), the five-session group showed a significant post-intervention increase (+11.6 points, p = 0.012), but no significant between-group change was observed.

To assess the subjective perception of change, we used linear mixed models to the selected CISMA items, where we included items 1, 3 and 4 for what we wanted to measure that was emotional changes, where time was incorporated as a covariate. so q Although tests of normality of residuals were significant, the diagnostic plots supported the adequacy of the model. None of the CISMA items showed statistically significant differences between groups after the intervention, and the models explained limited variance, although slight trends in favor of the five-session group were observed.

These results have been incorporated into the revised manuscript. The use of mixed models with covariate adjustment provides a more conservative and informative analysis framework that reduces the likelihood of spurious findings.

Reviewer 1 – Comment 6:

The authors also conducted a series of correlations between expectation levels and various outcomes. However, since expectation levels are not directly related to the intervention itself, this analysis falls outside the study’s scope and does not support the intervention’s effectiveness.

Response:

We appreciate the reviewer’s thoughtful observation regarding the correlation analyses between patients’ expectations and clinical outcomes. We fully agree that expectations represent a distinct construct from the intervention itself, and therefore, any associations identified between expectations and outcomes cannot be interpreted as direct evidence of the intervention’s effectiveness. As noted, significant correlations (e.g., p < 0.05) do not imply causality.

Furthermore, these correlations were not part of the study’s primary experimental design and did not account for key variables such as intervention group or time effects. Thus, they were presented as exploratory findings intended to complement the main results by highlighting possible psychosocial factors associated with changes in clinical scores (e.g., DASS-21), rather than as proof of treatment efficacy. We have clarified this point in the revised manuscript to avoid misinterpretation.

In response to the reviewer’s comment, we also conducted an additional, more robust analysis using linear mixed models to examine whether expectation level (neutral vs. positive vs. very positive) moderated changes in DASS-21 subscale scores over time, while also controlling for the diagnostic group (depression vs. bipolar disorder). These interaction effects were evaluated separately for the experimental and control groups.

Results indicated that expectation level did not significantly moderate the change in stress, anxiety, or depression scores in either group. However, in the experimental group (five-session condition), we observed non-significant trends suggesting greater symptom reduction among participants with very positive expectations, particularly on the stress and depression subscales. This pattern was not evident in the control group (one-session condition), which could suggest a potential interaction between high expectations and greater intervention intensity, though these findings remain inconclusive.

These moderation analyses and their corresponding estimates are now included in Tables 2 and 5 of the revised manuscript. We believe these additional results enrich the exploratory scope of the study while reinforcing a cautious interpretation of the role of expectations in therapeutic response.

Reviewer 1 – Comment 7:

Several wording issues should also be addressed:

1. The term “bilateral” significance level is not standard.

2. In the data analysis section, the authors describe a “descriptive analysis” that characterizes the sample, including measures of central tendency and dispersion (percentiles). However, dispersion should be measured by variance/standard deviation when using the mean and by interquartile range when using the median. These essential dispersion measures are missing from the results.

Response:

We thank the reviewer for this helpful comment.

1. We acknowledge that the term “bilateral” significance level is non-standard in English-language scientific writing. In the revised manuscript, we have replaced it with the appropriate term “significance level.”

2. Regarding the descriptive analysis, we agree that the measures of dispersion should align with the type of central tendency used: standard deviation or variance when reporting means, and interquartile range when reporting medians. In the updated version of the manuscript, we have included these essential dispersion measures in the Results section to provide a more complete and accurate summary of the sample characteristics. Specifically, we included means.

These revisions have been implemented to ensure clarity and adherence to statistical reporting standards.

Reviewer 2 – Comment 1: Literature review:

Key literature on music therapy within inpatient settings is missing, as is the unique inpatient context. The authors are encouraged to perform a search on pubmed or similar with terms "music therapy" "mental health" "inpatient" to summarise knowledge to date. See particularly work by Michael J. Silverman (where he has offered many different high intensity interventions), Catherine Carr (see systematic review of music therapy for acute adult psychiatric inpatients in PLOS ONE for an overview of practice and methods up to 2012 - https://pmc.ncbi.nlm.nih.gov/articles/PMC3732280/ and PhD thesis) and wider considerations of inpatient music therapy practice.

Response:

Thank you very much for this valuable suggestion. We have carefully reviewed the literature on music therapy in inpatient psychiatric settings, particularly the work of Michael J. Silverman and Catherine Carr, as well as relevant systematic reviews. In response to your comment, we have substantially expanded the Introduction section to better contextualize our study within current knowledge and practice.

Specifically, we incorporated the following key references into the revised manuscript:

1. Silverman MJ. Effects of music therapy on psychiatric patients’ proactive coping skills: Two pilot studies. Arts Psychother. 2011;38(2):125-9. doi:10.1016/j.aip.2011.02.004.

2. Silverman MJ. Effects of group songwriting on motivation and readiness for treatment on patients in detoxification: a randomized wait-list effectiveness study. J Music Ther. 2012;49(4):414-29. doi:10.1093/jmt/49.4.414.

3. Silverman MJ. The Influence of Music on the Symptoms of Psychosis: A Meta-Analysis Journal of Music Therapy, XL. 2003. Available from: http://jmt.oxfordjournals.org/

4. Carr C, Odell-Miller H, Priebe S. A Systematic Review of Music Therapy Practice and Outcomes with Acute Adult Psychiatric In-Patients. Vol. 8, PLoS ONE. 2013.

5. Gold C, Solli HP, Krüger V, Lie SA. Dose–response relationship in music therapy for people with serious mental disorders: systematic review and meta-analysis. Clin Psychol Rev. 2009;29(3):193-207. doi:10.1016/j.cpr.2009.01.001.

Importantly, our revised introduction now makes clear that this study directly addresses a gap in the literature: the limited evidence regarding the impact of short-term, high-intensity music therapy assited by MIDI during brief psychiatric hospitalizations. We have also included this point in the Discussion section when interpreting our findings.

We are grateful for your guidance, which helped us significantly improve the theoretical framing of our work.

Reviewer 2 – Comment 2: Procedures:

Timing of the outcome assessments is not clear. More generally, the procedure from recruitment into the study, randomisation, intervention delivery and follow-up needs greater clarity. The protocol published under ISRCTN provides an interesting chart suggesting patients were recruited in daily cohorts. Some explanation of how many groups ran (was it one each of high intensity and low intensity or several over the course of the study?) and at what point assessments were taken would be helpful. Please define the time window allowed to collect follow-up data.

Response:

We appreciate this important observation and have revised the manuscript to clarify the procedures regarding recruitment, randomisation, intervention delivery, and follow-up timing.

Patients were consecutively recruited upon admission to the psychiatric unit between August 1st and September 2nd. Inclusion was restricted to those who had been hospitalized for at least 3 to 5 days and showed sufficient clinical stability to participate in up to five music therapy sessions. Upon meeting eligibility criteria, patients were randomly assigned to either the high-frequency (5 sessions/week) or low-frequency (1 session/week) music therapy group.

New intervention cohorts were formed approximately every 4 days, depending on patient admission and availability. The DASS-21 and Satisfaction With Life Scale (SWLS) were administered at two time points:

● Baseline: Before the first music therapy session.

● Post-intervention: On the fifth day of participation, regardless of group assignment.

The CISMA scale was administered in a pre-post format during the first music therapy session for both groups. Additionally, for participants in the high-frequency group, the CISMA was also administered in a second pre-post format during the fifth and final session.

Follow-up data were collected within a fixed window of 5 days from the start of the intervention, and patients discharged before completing the post-intervention assessments were excluded from the final analysis.

Importantly, all participants remained in their originally assigned groups throughout the study, and no crossovers occurred. This ensured consistency in exposure to the intervention and the integrity of group comparisons.

These clarifications have been incorporated into the revised manuscript to provide a more complete and transparent description of the study timeline and methodology.

Reviewer 2 – Comment 3: Procedures:

Please provide a brief description of how each of the scales is rated and scored along with psychometric properties.

Response:

We appreciate the suggestion to expand the description of the instruments used in our study. We have included detailed information on the scoring and psychometric properties of each scale into the revised manuscript.

Reviewer 2 – Comment 4: Procedures:

Please provide more detail on your secondary analysis of patient expectations and the methods used to analyse this.

Response:

We appreciate the reviewer’s interest in the secondary analysis of patient expectations. As previously detailed in response to similar feedback from the first reviewer, we have conducted thorough analyses exploring the role of expectations using both correlation methods and more robust linear mixed models that account for intervention group, time, and diagnostic category. These analyses—presented in detail in Tables 5—investigated whether expectation levels moderated changes in clinical outcomes (DASS-21 subscales) over time.

While these exploratory findings suggest possible trends, particularly in the high-intensity intervention group, they do not provide conclusive evidence regarding the direct effect of expectations on treatment efficacy. We have clarified this distinction in the revised manuscript to avoid overinterpretation.

Reviewer 2 – Commen

---

## [Decision Letter · Decision Letter 1]

1 Dec 2025

Dear Dr. Ferro,

Unfortunately, one of the reviewers was no longer available, and despite considerable efforts, I was unable to recruit additional reviewers. I therefore ask for your understanding regarding the delay, and have now prepared the review myself. As a consequence, several new points have been added that I would ask the authors to address in a further revision (see below).

The abstract is not appropriately written. Please use the CONSORT extension for abstracts. Important information is missing, including details on allocation concealment and the timing of the intervention. One sentence (line 36/37) is still written in the future tense and should be revised. The results section of the abstract is too brief; numerical results rather than only p-values should be reported. Furthermore, the abstract should not present figures describing changes over time within groups, but should instead present between-group differences to make clear that—with one exception—no differences between the groups were found across all outcomes. This also needs to be reflected in the abstract’s discussion. At present the abstract appears to suggest positive effects, but as this is a randomized controlled trial and not a before-after–post study, this distinction must be made explicit. As in other parts of the manuscript, the DASS instrument is incorrectly abbreviated as “DAS” here as well.

The manuscript should state and explain why the trial registration was conducted post hoc. This is a major limitation that should be clearly emphasized. The manuscript also refers to a study protocol; however, registration is not a study protocol. It appears that no protocol exists, which must be stated. Please revise the manuscript in accordance with the current CONSORT 2025 Statement, which contains elements that have not yet been considered.

In the submitted CONSORT checklist, “not applicable” is listed under blinding. This is inappropriate. The lack of blinding should be described in the Methods section, not only in the Discussion.

The background section cites several older reviews and original studies. This should be revised to include more recent work, particularly the reviews by Aalbers and Zhong, which are already cited in the discussion. The section added in the revision describing Silverman’s studies (from line 116 onward) should be removed, as these studies were already included in the review by Carr mentioned before. The reference to Goldberg’s meta-analysis should also be removed. Although it is described as “recent evidence,” it was published in 2009 and is therefore outdated.

The final paragraph of the introduction (from line 134 onward) should be removed.

The Methods section is generally complete, aside from the missing information on blinding (see above). However, it requires more structure, and the CONSORT 2025 Statement can serve as a helpful guide for organizing the content and reducing redundancies, which currently occur frequently, for example, in the descriptions of randomization and allocation concealment, which are presented in varying detail across different sections.

The sample size calculation should be explicitly labelled as such, rather than described merely as “sample.”

The text beginning at line 167 does not belong in the Methods section, as it describes characteristics of the sample.

In line 196, the manuscript refers to “8 periods over 6 weeks.” However, the study was conducted only from 1 August to 2 September, so it is unclear how 6 weeks of data collection were possible.

As this is a complex intervention, the authors should consider presenting the intervention in accordance with the TIDieR checklist. If appropriate, the description of the intervention from line 313 onward could be expanded based on TIDieR.

The predefined primary endpoint is the DASS scale, which consists of three components. However, these three components were analyzed separately and only one of them showed significant differences. The authors should consider whether this constitutes multiple testing and whether a Bonferroni correction should be applied to the primary endpoint. With three primary endpoints, the adjusted p-value threshold would be 0.0125 instead of 0.05. At minimum, this issue should be critically reflected upon and discussed.

The effects need to be discussed much more cautiously given the numerous limitations. For example, line 560 suggests “a clear effect,” whereas the results more appropriately indicate “suggestive” evidence rather than clear effects.

The newly added discussion of the studies by Carr and Silverman (from line 644 onward) should be removed. I acknowledge that this was added in response to a previous review, but these studies are relatively old, and newer evidence is available; therefore, this section should be deleted.

A very important limitation that must be added is that participants who did not receive the intervention were excluded from the analysis. This means that the study did not conduct an intention-to-treat analysis but a per-protocol analysis. This is a major limitation that should be prominently highlighted.

Although the lack of blinding is mentioned as a limitation, the authors should also discuss whether it would have been possible to include additional, potentially more objective outcomes, such as length of stay, effectiveness of psychiatric treatment, or medication use. These could at least have been included as secondary outcomes to reduce the risk of bias associated with the lack of blinding.

We look forward to receiving your revised manuscript.

Kind regards,

Sascha Köpke

Academic Editor

PLOS ONE

Journal Requirements:

Reviewers' comments:

Reviewer's Responses to Questions

**Comments to the Author**

Reviewer #1: (No Response)

2. Is the manuscript technically sound, and do the data support the conclusions?

Reviewer #1: No

3. Has the statistical analysis been performed appropriately and rigorously?

Reviewer #1: No

4. Have the authors made all data underlying the findings in their manuscript fully available?

Reviewer #1: Yes

5. Is the manuscript presented in an intelligible fashion and written in standard English?

Reviewer #1: Yes

Reviewer #1: The authors have made very substantial changes in response to the reviewers’ concerns. However, the following issues require further clarifications.

1. The authors report a sample size of n = 30 per group based on the specifications (α = 0.05, power = 0.8, Cohen’s *d* = 0.6, group ratio = 1:1). However, my calculations using both G*Power and R’s pwr package (function: pwr.t.test()) yield different estimates:

• Two-tailed test: n = 45 per group.

• One-tailed test: n = 36 per group.

Could the authors check their sample size estimate and also clarify whether a one- or two-tailed test was assumed?

2. I raised the issue of not using repeated measures analysis in my previous round of comments. I appreciate the authors taking my advise and revised their entire analysis. However, this change introduces an inconsistency between the sample size justification and the actual analytical model – the sample size is justified based on a simple independent sample t-test while the analysis is a much more complicated linear mixed model. I understand that power analysis for linear mixed model is more complex. I recommend that the authors to acknowledge this discrepancy and comment briefly on potential power limitation. A post hoc power analysis may also be considered.

3. Additionally, the authors stated that the model diagnostics supported the use of linear mixed model due to homoscedastic residuals and the absence of influential outlier. Can the authors provide supports of these statements in the supplementary documents?

4. Only three items from the CISMA instrument were used. Why were only these three items selected? Additionally, is was stated that the CISMA has shown strong content validity and responsiveness in clinical music therapy practice. Any supports for this statement?

5. A single item was used for measuring overall SWLS. It was stated that “factor analysis also supports its reliability with communalities of 0.70 and 0.77”. How was the factor analysis conducted with only one item? Also, what does “corrected attenuation” mean? It is also strange to argue for construct validity for a single item. These statements suggest that the authors may have some misconceptions of psychometric properties.

6. The authors did not adequately address the issue of multiple comparison bias raised in my previous comments. While a linear mixed model was used, this does not inherently correct for the increased risk of false positives when testing multiple outcome variables. If a Bonferroni correction (adjusting for four comparisons) were applied, the significance threshold would be reduced to p < 0.0125 (0.05/4), rendering the reported findings non-significant. This underscores the need for caution in interpreting the results, as they may be susceptible to Type I error inflation.

**Do you want your identity to be public for this peer review?** For information about this choice, including consent withdrawal, please see our For information about this choice, including consent withdrawal, please see our Privacy Policy .

Reviewer #1: No

---

## [Author Response · Author response to Decision Letter 2]

19 Jan 2026

We would like to thank the Academic Editor and the reviewer for their careful evaluation of our manuscript entitled “High-Frequency vs. Low-Frequency MIDI-assisted Group Music Therapy in Psychiatric Inpatients: A Randomized Controlled Trial” (Manuscript ID: PONE-D-24-55131R1).

We are grateful for the detailed and constructive comments, which have been invaluable in improving the clarity, methodological rigor, and overall quality of the manuscript. We have revised the manuscript extensively in response to all points raised.

Below, we provide a point by point response to each comment from the Academic Editor and the reviewer. A marked-up version of the revised manuscript with track changes enabled has been submitted as “Revised Manuscript with Track Changes”, allowing all modifications to be easily identified. In addition, a clean version of the revised manuscript, without tracked changes, has been submitted as “Manuscript”.

Reviewer 1 – Comment 1:

The abstract is not appropriately written. Please use the CONSORT extension for abstracts. Important information is missing, including details on allocation concealment and the timing of the intervention. One sentence (line 36/37) is still written in the future tense and should be revised. The results section of the abstract is too brief; numerical results rather than only p-values should be reported.

Furthermore, the abstract should not present figures describing changes over time within groups, but should instead present between-group differences to make clear that—with one exception—no differences between the groups were found across all outcomes. This also needs to be reflected in the abstract’s discussion. At present the abstract appears to suggest positive effects, but as this is a randomized controlled trial and not a before-after–post study, this distinction must be made explicit. As in other parts of the manuscript, the DASS instrument is incorrectly abbreviated as “DAS” here as well.

Response:

We thank the reviewer for this detailed and constructive feedback. The abstract has been thoroughly revised in accordance with the CONSORT extension for abstracts. Specifically, we have clarified the randomized controlled trial design, explicitly described allocation concealment and the timing of the intervention, and corrected the abbreviation of the Depression Anxiety Stress Scale to DASS-21 throughout the manuscript.

The Results section of the abstract has been expanded to report numerical between group estimates rather than p-values alone, and the focus has been shifted from within group changes to between group differences, as appropriate for a randomized controlled trial. The wording of the Conclusions has also been revised to avoid a before and after interpretation and to clearly state that, with the exception of stress symptoms, no significant between-group differences were observed across outcomes. All future-tense phrasing has been corrected.

We believe these revisions substantially improve the clarity, methodological accuracy, and CONSORT compliance of the abstract.

Reviewer 1 – Comment 2:

The manuscript should state and explain why the trial registration was conducted post hoc. This is a major limitation that should be clearly emphasized. The manuscript also refers to a study protocol; however, registration is not a study protocol. It appears that no protocol exists, which must be stated.

Response:

We thank the reviewer for these related comments and agree that clearer distinctions were needed between trial registration and the existence of a study protocol. We have revised the manuscript to explicitly state that the trial was registered retrospectively, which constitutes a major methodological limitation, and that no publicly registered protocol exists. We now clarify that trial registration and protocol development are distinct processes, and that retrospective registration does not replace a prospectively registered protocol.

We further explain that, within the Colombian regulatory context, prospective registration of clinical trials on international platforms is not a mandatory requirement. However, we emphasize that this regulatory context does not mitigate the methodological limitation associated with retrospective registration, which restricts public verification of the study timeline and pre-specified methods.

At the same time, we clarify that the study was conducted under a complete written internal protocol developed prior to data collection, which specified the study design, intervention procedures, outcomes, and analysis plan, and which was reviewed and approved by an independent Ethics Committee in accordance with national regulations. Nevertheless, we now explicitly acknowledge that this protocol was not publicly accessible.

These revisions were made to improve conceptual clarity and transparency and to align the manuscript with CONSORT recommendations.

Reviewer 1 – Comment 3:

Please revise the manuscript in accordance with the current CONSORT 2025 Statement, which contains elements that have not yet been considered. In the submitted CONSORT checklist, “not applicable” is listed under blinding. This is inappropriate. The lack of blinding should be described in the Methods section, not only in the Discussion.

Response:

We thank the reviewer for this important comment. The manuscript has been revised in accordance with the CONSORT 2025 Statement, and the reporting of methodological details has been updated accordingly.

Specifically, the issue of blinding has now been explicitly addressed in the Methods section. A dedicated subsection (“Blinding”) was added to clarify that blinding of participants and therapists was not feasible due to the nature of the music therapy intervention. We now clearly describe that participants and the music therapist were aware of group allocation, while participant recruitment and enrollment were conducted by a coordinator blinded to the randomization sequence, which was generated independently and not accessible during enrollment. We also clarify that outcomes were assessed using self-reported questionnaires and that no formal blinding of outcome assessment was implemented.

In addition, the CONSORT checklist has been corrected to remove “not applicable” under blinding and now accurately reflects the procedures described in the Methods section. These revisions were made to improve transparency and ensure full compliance with CONSORT 2025 reporting standards.

Reviewer 1 – Comment 4:

The background section cites several older reviews and original studies. This should be revised to include more recent work, particularly the reviews by Aalbers and Zhong, which are already cited in the discussion. The section added in the revision describing Silverman’s studies (from line 116 onward) should be removed, as these studies were already included in the review by Carr mentioned before. The reference to Goldberg’s meta-analysis should also be removed. Although it is described as “recent evidence,” it was published in 2009 and is therefore outdated.

Response:

We have removed the older reviews and original studies identified by the reviewer from the Background section. Specifically, the section describing Silverman’s individual studies has been deleted, as these studies were already synthesized within previously cited reviews, and the reference to Goldberg’s meta-analysis (2009) has also been removed.

The Discussion has been updated to reflect more recent evidence, with citations primarily from 2020 onward, which now frame the interpretation of the findings.

Reviewer 1 – Comment 5:

The final paragraph of the introduction (from line 134 onward) should be removed.

Response:

The final paragraph of the Introduction (from line 134 onward) has been removed in the revised manuscript, as requested.

Reviewer 1 – Comment 6:

The Methods section is generally complete, aside from the missing information on blinding (see above). However, it requires more structure, and the CONSORT 2025 Statement can serve as a helpful guide for organizing the content and reducing redundancies, which currently occur frequently, for example, in the descriptions of randomization and allocation concealment, which are presented in varying detail across different sections.

Response:

The Methods section has been restructured in accordance with the CONSORT 2025 Statement. Redundant descriptions of randomization and allocation concealment have been removed and consolidated into a single dedicated subsection. Information on blinding has been clarified and organized following CONSORT recommendations. Overall, the section was reorganized to improve clarity, reduce repetition, and enhance transparency.

Reviewer 1 – Comment 7:

The sample size calculation should be explicitly labelled as such, rather than described merely as “sample.”

Response:

Thank you for this comment. We have clarified the manuscript by explicitly labelling the subsection as “Sample size calculation”, in accordance with CONSORT recommendations. In addition, we clarified in the Participants section that the description of the inpatient population refers exclusively to the study setting and eligibility context, and not to the sample size estimation procedure. This distinction was made to avoid ambiguity between the a priori statistical calculation and the characteristics of the enrolled participants.

Reviewer 1 – Comment 8:

The text beginning at line 167 does not belong in the Methods section, as it describes characteristics of the sample.

Response:

The text beginning at line 167 has been removed from the Methods section and relocated to the Results section, where sample characteristics are reported.

Reviewer 1 – Comment 9:

In line 196, the manuscript refers to “8 periods over 6 weeks.” However, the study was conducted only from 1 August to 2 September, so it is unclear how 6 weeks of data collection were possible.

Response:

The wording in line 196 has been revised to avoid ambiguity. The text “8 periods over 6 weeks” referred to overlapping time periods of group cohorts recruited. The reference to “six weeks” has been removed, and the text now clarifies that the eight periods correspond to sequential recruitment blocks formed approximately every four days during the enrollment phase (1 August to 2 September), rather than to follow-up or observation periods.

Reviewer 1 – Comment 10:

As this is a complex intervention, the authors should consider presenting the intervention in accordance with the TIDieR checklist. If appropriate, the description of the intervention from line 313 onward could be expanded based on TIDieR.

Response:

We made a supplementary material that provides a structured and detailed description of the intervention components, procedures, materials, delivery, and dosage, following the TIDieR framework. However, it is too long to inclue it in the manuscript text. In the manuscript, we described the intervention in accordance with the TIDieR checklist, adapted to make it easy for the readers. We have now explicitly referenced this appendix in the Methods section to improve clarity and accessibility for readers (page 13 of the final manuscript).

Reviewer 1 – Comment 11:

The predefined primary endpoint is the DASS scale, which consists of three components. However, these three components were analyzed separately, and only one of them showed significant differences. The authors should consider whether this constitutes multiple testing and whether a Bonferroni correction should be applied to the primary endpoint. With three primary endpoints, the adjusted p-value threshold would be 0.0125 instead of 0.05. At minimum, this issue should be critically reflected upon and discussed.

Response:

We agree that the analysis of the three DASS-21 subscales raises the issue of multiplicity if they are interpreted as independent primary endpoints.

In this trial, the DASS-21 was defined as a multidimensional primary outcome, reflecting a single overarching construct of psychological distress operationalized through three conceptually and statistically interrelated subscales (depression, anxiety, and stress). Accordingly, the subscales were treated as correlated coprimary endpoints, which is common practice in clinical trials using multidimensional psychometric instruments.

Bonferroni correction assumes statistical independence among tests and is known to be overly conservative when applied to correlated measures derived from the same instrument. Given the moderate-to-high intercorrelations among DASS-21 subscales consistently reported in the literature, applying a Bonferroni-adjusted threshold would substantially increase the risk of Type II error and potentially obscure clinically meaningful effects. In addition, the use of linear mixed-effects models partially accounts for shared variance across repeated measures.

Nonetheless, we acknowledge that the analysis of multiple subscales entails a risk of multiplicity. In response to the reviewer’s comment, we have explicitly addressed this issue in the Discussion, emphasizing that the between-group interaction observed for the stress subscale should be interpreted cautiously and as suggestive rather than definitive evidence, particularly in light of multiple testing and other methodological limitations.

Reviewer 1 – Comment 12:

The effects need to be discussed much more cautiously given the numerous limitations. For example, line 560 suggests “a clear effect,” whereas the results more appropriately indicate “suggestive” evidence rather than clear effects.

Response:

We agree with the reviewer that the interpretation of the findings required a more cautious tone given the study’s methodological limitations. In response, we have revised the Discussion section to remove language implying clear or definitive effects, including the statement previously referring to a “clear effect.” The findings are now consistently described as suggestive and exploratory, and the conclusions have been aligned with the acknowledged limitations of the study. These revisions ensure that the interpretation is proportional to the strength of the evidence and consistent with CONSORT recommendations.

Reviewer 1 – Comment 13:

The newly added discussion of the studies by Carr and Silverman (from line 644 onward) should be removed. I acknowledge that this was added in response to a previous review, but these studies are relatively old, and newer evidence is available; therefore, this section should be deleted.

Response:

In response, we have removed the discussion of the studies by Carr and Silverman from the manuscript, as recommended. This section has been deleted in its entirety.

To ensure that the Discussion remains grounded in the most current evidence, we have revised the relevant paragraphs and incorporated more recent systematic reviews, meta-analyses, and controlled studies addressing the effects of music therapy in acute psychiatric settings and the role of intervention intensity. These updates allow the Discussion to reflect contemporary evidence while maintaining coherence with the study findings.

Reviewer 1 – Comment 14:

A very important limitation that must be added is that participants who did not receive the intervention were excluded from the analysis. This means that the study did not conduct an intention-to-treat analysis but a per-protocol analysis. This is a major limitation that should be prominently highlighted.

Response:

We agree that the analytic approach corresponds to a per-protocol framework, because participants who did not receive any intervention were excluded from the final analysis. However, this decision is directly aligned with the original design and scientific objective of the study, which from the title, aim, and protocol stage focused specifically on comparing two active intervention conditions: high-frequency versus low-frequency music therapy.

Participants who did not attend any session could not be meaningfully classified into either intensity group, and retaining them in an intention-to-treat analysis would have introduced bias in estimating the dose–response relationship inherent to the primary research question. Nonetheless, we fully acknowledge that analyzing only participants who received the allocated intervent

---

## [Editor Report · Decision Letter 2]

1 Feb 2026

Dear Dr. Ferro,

We look forward to receiving your revised manuscript.

Kind regards,

Sascha Köpke

Academic Editor

PLOS One

---

## [Author Response · Author response to Decision Letter 3]

19 Feb 2026

Academic editor – Comment 1:

The information we provided in the ‘Funding Information’ and ‘Financial Disclosure’ sections does not match.

Response:

We revised the information we provided in the ‘Funding Information’ and ‘Financial Disclosure’.

We want to highlight that this study did not receive any external grant funding or any financial support from any institution. We recognized the health institution (Clinica Montserrat – Hospital Universitario) with an acknowledgement for their support and facilities, but we did not receive any financial support. Only the authors had a role in the study design, data collection and analysis, decision to publish, and preparation of the manuscript. The authors covered all project costs.

Also, we included in the cover letter the following “Funding disclosure statement”:

“This study did not receive any external grant funding. The authors covered all project costs. No institution had any role in the study design, data collection and analysis, decision to publish, or preparation of the manuscript. The authors declare that they have no financial or non-financial competing interests.”

Sincerely,

Eugenio Ferro, MD.

Corresponding author

eferro@unbosque.edu.co

---

## [Editor Report · Decision Letter 3]

24 Feb 2026

High-Frequency vs. Low-Frequency MIDI-assisted Group Music Therapy in Psychiatric Inpatients: A Randomized Controlled Trial

PONE-D-24-55131R3

Dear Dr. Ferro,

We’re pleased to inform you that your manuscript has been judged scientifically suitable for publication and will be formally accepted for publication once it meets all outstanding technical requirements.

Kind regards,

Sascha Köpke

Academic Editor

PLOS One
---

## [Editor Report · Acceptance letter]

PONE-D-24-55131R3

PLOS One

Dear Dr. Ferro,

I'm pleased to inform you that your manuscript has been deemed suitable for publication in PLOS One. Congratulations! Your manuscript is now being handed over to our production team.

Kind regards,

on behalf of

Professor Sascha Köpke

Academic Editor

PLOS One